# Acute Kidney Injury in Patients with Severe ARDS Requiring Extracorporeal Membrane Oxygenation: Incidence, Prognostic Impact and Risk Factors

**DOI:** 10.3390/jcm11041079

**Published:** 2022-02-18

**Authors:** Kevin Pilarczyk, Katharina Huenges, Burkhard Bewig, Lorenz Balke, Jochen Cremer, Assad Haneya, Bernd Panholzer

**Affiliations:** 1Department of Intensive Care Medicine, Imland Klinik Rendsburg, 24768 Rendsburg, Germany; 2Department of Cardiovascular Surgery, University of Schleswig-Holstein, 24105 Kiel, Germany; katharina.huenges@uksh.de (K.H.); jochen.cremer@uksh.de (J.C.); assad.haneya@uksh.de (A.H.); bernd.panholzer@uksh.de (B.P.); 3Department of Pneumology, Städtisches Krankenhaus Kiel, 24116 Kiel, Germany; burkhard.bewig@krankenhaus-kiel.de (B.B.); lorenz.balke@krankenhaus-kiel.de (L.B.)

**Keywords:** acute respiratory distress syndrome, extracorporeal membrane oxygenation, acute kidney injury

## Abstract

(1) Background: Acute kidney injury (AKI) is a common but under-investigated complication in patients receiving extracorporeal membrane oxygenation (ECMO). We aimed to define the incidence and clinical course, as well as the predictors of AKI in adults receiving ECMO support. (2) Materials and Methods: This is a retrospective analysis of all patients undergoing veno-venous ECMO treatment in a tertiary care center between December 2008 and December 2017. The primary endpoint was the new occurrence of an AKI of stage 2 or 3 according to the Kidney Disease: Improving Global Outcomes (KDIGO) classification after ECMO implantation. (3) Results: During the observation period, 103 patients underwent veno-venous ECMO implantation. In total, 59 patients (57.3%) met the primary endpoint with an AKI of stage 2 or 3 and 55 patients (53.4%) required renal replacement therapy. Patients with an AKI of 2 or 3 suffered from more bleeding and infectious complications. Whereas weaning failure from ECMO (30/59 (50.8%) vs. 15/44 (34.1%), *p* = 0.08) and 30-day mortality (35/59 (59.3%) vs. 17/44 (38.6%), *p* = 0.06) only tended to be higher in the group with an AKI of stage 2 or 3, long-term survival of up to five years was significantly lower in the group with an AKI of stage 2 or 3 (*p* = 0.015). High lactate, serum creatinine, and ECMO pump-speed levels, and low platelets, a low base excess, and a low hematocrit level before ECMO were independent predictors of moderate to severe AKI. Primary hypercapnic acidosis was more common in AKI non-survivors (12 (32.4%) vs. 0 (0.0%), *p* < 0.01). Accordingly, pCO_2_-levels prior to ECMO implantation tended to be higher in AKI non-survivors (76.12 ± 27.90 mmHg vs. 64.44 ± 44.31 mmHg, *p* = 0.08). In addition, the duration of mechanical ventilation prior to ECMO-implantation tended to be longer (91.14 ± 108.16 h vs. 75.90 ± 86.81 h, *p* = 0.078), while serum creatinine (180.92 ± 115.72 mmol/L vs. 124.95 ± 77.77 mmol/L, *p* = 0.03) and bicarbonate levels were significantly higher in non-survivors (28.22 ± 8.44 mmol/L vs. 23.36 ± 4.19 mmol/L, *p* = 0.04). (4) Conclusion: Two-thirds of adult patients receiving ECMO suffered from moderate to severe AKI, with a significantly increased morbidity and long-term mortality.

## 1. Introduction

Since its introduction in 1972 by Hill et al., extracorporeal membrane oxygenation (ECMO) represents a mainstay of critical care for the treatment of adult patients with acute respiratory distress syndrome (ARDS) [1]. Although ECMO enables the efficient oxygenation and elimination of carbon dioxide, and studies such as the randomized controlled trial of Conventional ventilatory support vs extracorporeal membrane oxygenation for severe adult respiratory failure (CESAR) trial or the ECMO to rescue lung injury in severe ARDS (EOLIA) trial suggest a potential survival benefit, somewhat perplexingly, the outcomes of patients with ECMO support remains quite poor [2]. This might be at least partly explained by the invasive character of ECMO, with a high incidence of complications [3]. Acute kidney injury (AKI) is common in patients undergoing ECMO, with an incidence as high as 70–85%, and is associated with increased mortality rates of up to 80% [4]. AKI is present both before and after the initiation of ECMO. The underlying mechanisms for AKI among patients requiring ECMO appear to be complex, multifactorial, time-dependent and include prolonged prehypoxemia and hypercapnia, hypoperfusion, ischemia–reperfusion injury, inflammatory responses, coagulation–platelet abnormalities, and immune-mediated injury that arises from the primary underlying disease, premorbid conditions and the ECMO circuit [5]. Because AKI is the most common complication and is a major risk factor of mortality, defining the risk factors for AKI in these patients is extremely important. However, clarity with regard to the timing and risk factors for this renal insult is lacking. The available studies are mainly based on pediatric data and specific patient populations, respectively, or include a high percentage of patients requiring cardiovascular support. Against this background, we performed this retrospective cohort study to explore the incidence and outcome of and the risk factors for AKI (before and immediately after ECMO initiation) in adult patients receiving ECMO support for ARDS. In addition, we compared patients with AKI who could be discharged home to those who died during their hospital stay.

## 2. Materials and Methods

### 2.1. Patients

In total, 140 consecutive patients (minimum age: 18 years) undergoing veno-venous ECMO treatment for severe ARDS between December 2008 and December 2017 were screened for inclusion in this retrospective study. We used the standards for the reporting of diagnostic accuracy (STARD) statement for planning and conducting the study and for preparing the manuscript. A total of 24 patients with pre-ECMO AKI were excluded. In addition, 11 patients who had to be switched to or from veno-arterial/veno-veno-arterial-ECMO due to circulatory failure were excluded, since the pathophysiology and incidence of AKI vary between different modes of ECMO support. We also excluded 2 patients who survived for less than 24 h after ECMO initiation.

### 2.2. Ethics Principles

The present study was conducted according to the principles expressed in the Declaration of Helsinki and was approved by the Institutional Ethics Review Board of the university hospital of Kiel. Written informed consent from the patient or the patient’s next of kin was obtained.

### 2.3. Initiation and Management of ECMO

Patient selection, medical management, and the settings of the mechanical ventilation and ECMO circuits followed an institutional protocol. Briefly, patients with deteriorating hypoxemia (PaO_2_/FiO_2_ ratio of <80 on FiO_2_ of >90%) or uncompensated hypercapnia (CO_2_ retention with a pH < 7.20 despite a plateau pressure of >30 cmH_2_O) under advanced mechanical ventilator support with or without adjunctive therapies were considered for ECMO support. The final decisions regarding ECMO initiation were made after consultation with a multidisciplinary ECMO team consisting of intensivists, pulmonologists, and cardiothoracic surgeons. During the observation period, five different ECMO systems were used: (1) Cardiohelp^®^ (Getinge Group, Göteburg, Sweden), (2) Deltastream^®^ (Medos, Stolberg, Germany), (3) Deltastream DP2^®^ (Medos, Stolberg, Germany), (4) Permanent Life Support (PLS) System^®^ (Getinge Group, Göteburg, Sweden), (5) Novalung/iLA activve ^®^ (Novalung, Hechingen, Germany).

Implantation took place in the ICU. After sterile draping, the ECMO implant was initiated following the administration of a heparin bolus (100 IU/kg). Depending on body weight and size, venous drainage was performed by a long, heparin-coated 19, 23, 25 or 29 French (F) cannula. The cannula was placed percutaneously via the right femoral vein (VFC) in the inferior vena cava (IVC), just below its inflow into the right atrium. The venous cannulas were implanted under TEE guidance, using the bicaval view. For afferent access, the right internal JV was punctured, and a heparin-coated 15, 17 or 19 French arterial cannula was advanced so that its tip was located at the SVC–right atrial (RA) junction.

Pump blood flow and sweep gas flow rates were adjusted, to maintain target oxygen saturation and carbon dioxide removal rate at all times. During ECMO support, a pressure-controlled ventilation mode was used for patients trying to achieve lung-protective ventilation (FiO_2_ lower than 30%, a respiratory rate lower than 10–12 per minute, a positive end-expiratory airway pressure of more than 10 cmH_2_O, and a peak inspiratory pressure of 20–25 cmH_2_O to achieve low tidal ventilation (<5 mL/kg of predicted body weight)) and prevent ventilator-induced lung injury. If patients were stable and tolerant of treatment, we changed the mode of the MV to pressure support ventilation and subsequently adjusted the ventilator settings for weaning. The possibility of weaning off from ECMO was assessed daily, and an off-support trial was performed to determine decannulation when arterial blood gas was maintained within the target range, with a sweep gas flow of 1 L/min or less, regardless of pump blood flow at acceptable ventilator settings. Patients who maintained adequate gas exchange without sweep gas flow (sweep gas-off trial) were closely monitored for at least 2 h, and decannulation was considered for patients who were stable during this period. The decisions about the total duration of the weaning trial and decannulation were made by the intensivists treating the patient and the ECMO team.

### 2.4. Definition of Endpoint and Outcomes

The AKI stage was determined daily based on diuresis rate and serum creatinine concentration, according to the “Kidney Disease: Improving Global Outcomes” classification (KDIGO) as follows [6]:AKI 1: Increase of serum creatinine by ≥0.3 mg/dL (≥26.4 µmol/L) or increase to ≥150–200% from baseline or urine output < 0.5 mL/kg/h for >6 h;AKI 2: increase of serum creatinine to >200–300% from baseline and/or urine output < 0.5 mL/kg/h for >12 h;AKI 3: increase of serum creatinine to >300% from baseline or serum creatinine ≥ 4.0 mg/dL (≥354 µmol/L) after a rise of at least 44 µmol/L or treatment with renal replacement therapy and/or urine output < 0.3 mL/kg/h for > 24 h or anuria for 12 h.

The primary endpoint was the new occurrence of acute kidney injury at stage 2 or 3 after ECMO implantation. We chose to assess the risk of moderate to severe AKI rather than all AKIs because this severity (corresponding to KDIGO stage 2 and 3) has been shown to be associated with a significantly increased incidence of clinically important outcomes, such as the need for renal replacement therapy, death in hospital, and persistent renal dysfunction. The indication for RRT was made by the consultant in charge of the ICU at the time. RRT was initiated when severe AKI occurred along with any of the following that cannot otherwise be controlled: fluid overload, hyperkalemia, metabolic acidosis, or uremic symptoms. All patients were treated using continuous veno-venous hemodialysis. Replacement fluid was delivered into the extracorporeal circuit before the filter (predilution), with a ratio of dialysate to replacement fluid of 1:1. The effluent flow prescribed was based on the patient’s body weight and averaged 25–30 mL/kg/h. Blood flow was kept above 120 mL/min. Regional anticoagulation with citrate or systemic anticoagulation with intravenous heparin was used to prevent circuit clotting. Within the study period, two systems for hemodialysis were used: (1) Genius^®^ 90 (Fresenius Medical Care AG & Co., Bad Homburg vor der Höhe, Germany) and (2) multiFiltrate (Fresenius Medical Care AG & Co., Bad Homburg vor der Höhe, Germany).

Neurological complications were defined as a new occurrence of ischemic stroke, intracranial bleeding, or seizures.

Cardiac complications included myocardial infarction, as defined by the Fourth Universal Definition of Myocardial Infarction of the European Society Cardiology Clinical Practice Guidelines, cardiac arrest, and low cardiac output syndrome and significant arrhythmia.

### 2.5. Statistical Analysis

Statistical analyses were performed with SPSS Statistics 19 (IBM, Chicago, IL, USA). Continuous data were expressed as mean ± SD; categorical data were expressed as a percentage. Comparisons between two groups were carried out using an unpaired Student’s t-test for normally distributed data, or the Mann–Whitney rank-sum test for non-normally distributed data. Univariable analysis was performed on the quantitative variables using Student’s t-test or a Mann–Whitney test, and on the qualitative variables using the chi-square test or Fisher’s exact test. All variables showing a *p*-value of less than 0.1 between the two groups using Student’s t-test, the Mann–Whitney test, the chi-square test or Fisher’s exact test were selected for univariable analysis. Variables with *p* < 0.1 in the univariable analysis were included in a logistic regression model with backward selection, to determine the independent factors associated with AKI. Model calibration was tested using the Hosmer–Lemeshow test. Kaplan–Meier survival analysis was performed to estimate the early and long-term survival after ECMO implantation (Breslow (generalized Wilcoxon) test and log-rank (Mantel–Cox) test). Statistical significance was assumed for a *p*-value of <0.05.

## 3. Results

### 3.1. Patients’ Characteristics

After excluding 37 patients, 103 patients undergoing ECMO implantation due to severe ARDS were included in this study (see Figure 1). ECMO implantation was performed after an ICU stay of 4.99 ± 11.47 days. An AKI of stage 2 or 3 developed in 59 patients (57.3%), whereas 44 developed no or only mild AKI (42.7%). 55 patients (53.4%) required renal replacement therapy (RRT). AKI occurred 5.53 ± 8.2 days after ECMO implantation. Of those patients with AKI, 37 patients (62.7%) died during their hospital stay.

### 3.2. Comparison of Pre-ECMO Characteristics between AKI 2/3 and AKI 0/1

Baseline demographics, including age, gender, cause, and severity of ARDS (lung injury severity score), the duration of mechanical ventilation, and ventilator settings prior to ECMO initiation did not differ between the two groups of patients, either with AKI or without AKI (see Table 1).

The most common primary cause of ARDS was pneumonia in both groups. Patients with moderate to severe AKI were characterized by higher lactate levels (2.73 ± 2.40 vs. 1.58 ± 1.29 mmol/L, *p* = 0.008) and a lower base excess level (−1.70 ± 7.56 vs. 2.36 ± 5.43, *p* = 0.007) prior to ECMO implantation, indicating a higher severity of illness with more severe hypoperfusion. In addition, renal function assessed by serum creatinine (144.94 ± 95.95 vs. 83.19 ± 30.11 mmol/L, *p* < 0.001), as well as urea (13.08 ± 8.43 vs. 7.85 ± 5.20 mmol/L, *p* < 0.001), was impaired in patients developing AKI stage 2 or 3 (AKI 2/3). Whereas the distribution of relevant comorbidities, including COPD, CAD, or PHT, were comparable between groups, the prevalence of diabetes (15/59 (25.9%) vs. 4/44 (9.1%), *p* = 0.043) and chronic kidney disease (15/59 (25.4%) vs. 1/44 (2.3%), *p* < 0.001) was higher in the AKI 2/3 group. The type of ECMO system or implantation year was not associated with AKI incidence (not shown in the table).

In contrast to the initial blood and airflow of the ECMO system, the initial pump speed was significantly higher in the AKI 2/3 group (3370.53 ± 472.59 vs. 2970.55 ± 348.33/min, *p* = 0.035).

### 3.3. Impact on Outcome

Patients with AKI 2/3 were characterized by a more complicated course of illness, indicated by a higher incidence of pulmonary (17/59 (28.8%) vs. 5/44 (11.4%), *p* = 0.033) and ENT bleeding complications (47/59 (79.7%) vs. 22/44 (50.0%), *p* = 0.008) (Table 2. In addition, infectious complications (56/59 (94.9%) vs. 35/44 (79.5%), *p* < 0.001), including positive blood cultures (33/59 (55.9%) vs. 12/44 (27.2%), *p* = 0.005), could be observed more frequently in the AKI 2/3 group.

Whereas weaning failure from ECMO (30/59 (50.8%) vs. 15/44 (34.1%), *p* = 0.08) and 30-day mortality (35/59 (59.3%) vs. 17/44 (38.6%), *p* = 0.06) only tended to be higher in the AKI 2/3 group, hospital mortality was significantly higher in the AKI 2/3 group (37/59 (62.7%) vs. 17/44 (38.6%), *p* = 0.021). In addition, we performed Kaplan–Meier survival analysis based on AKI severity. Moderate to severe AKI was significantly associated with long-term mortality up to five years after ECMO therapy (log-rank test *p* = 0.015, Breslow test *p* = 0.01, Figure 2). The Cox proportional hazard ratio analysis showed that moderate to severe AKI was significantly associated with 180-day mortality (HR 0.287 (95% CI 0.120–0.689); *p* = 0.005).

### 3.4. Predictors of AKI 2/3

Using multivariate logistic regression analysis, lactate pre-implant, base excess pre-implant, serum creatinine pre-implant, thrombocytes pre-implant, hematocrit pre-implant and initial ECMO pump speed were identified as independent predictors of AKI stage 2/3 (Table 3).

### 3.5. Comparison of Pre-ECMO Characteristics between AKI Survivors and AKI Non-Survivors

Baseline demographics, including age, gender, the cause and severity of ARDS (lung injury severity score), and ventilator settings prior to ECMO initiation did not differ between AKI survivors and AKI non-survivors (see Table 4). In contrast, primary hypercapnic acidosis was more common in non-survivors (12 (32.4%) vs. 0 (0.0%), *p* < 0.01). Accordingly, pCO_2_-levels prior to ECMO implantation tended to be higher in non-survivors (76.12 ± 27.90 mmHg vs. 64.44 ± 44.31 mmHg, *p* = 0.08) In addition, the duration of mechanical ventilation prior to ECMO implantation tended to be longer (91.14 ± 108.16 h vs. 75.90 ± 86.81 h, *p* = 0.078), while serum-creatinine (180.92 ± 115.72 mmol/L vs. 124.95 ± 77.77 mmol/L, *p* =0.03) and bicarbonate were significantly higher in non-survivors (28.22 ± 8.44 mmol/L vs. 23.36 ± 4.19 mmol/L, *p* = 0.04).

### 3.6. Complications after ECMO Implantation in AKI Survivors Compared to AKI Non-Survivors

Whereas cardiac complications (16/37 (43.2%) vs. 3/22 (13.6%), *p* = 0.0186), as well as infectious complications (37/37 (100.0%) vs. 19/22 (86.4%), *p* = 0.047), could be observed more frequently in AKI non-survivors, the incidence of neurological complications was significantly higher in AKI survivors (11/22 (50.0%) vs. 8/37 (21.6%) *p* = 0.024) (see Table 5). Out of 37 AKI non-survivors, seven patients could be weaned successfully from ECMO, whereas 30 patients died during extracorporeal support.

## 4. Discussion

### 4.1. Introduction

ECMO is considered a therapeutic option for patients who have severe ARDS with refractory hypoxemia or who are unable to tolerate volume-limited strategies [1,2]. The use of this technique has been growing exponentially in the last decade, encouraged by promising results from the multi-centered, randomized controlled trial, CESAR, and those benefits described during the influenza A(H1N1) pandemic [2]. However, ECMO is still marred by a high rate of complications such as bleeding, thrombosis, and nosocomial infection, being associated with a significant increase in morbidity and mortality [3]. AKI is frequently reported and related to poor prognosis and high mortality rates [4,7,8,9,10]. However, clarity with regard to the incidence and timing for this renal insult is lacking, and studies investigating the pathophysiological features and risk factors of AKI are rare. Moreover, the available studies are mainly based on pediatric data or include lung as well as cardiovascular support. Therefore, adult data about incidence and prognostic impact, especially the risk factors and predictors of AKI in patients with veno-venous lung support for severe ARDS, are insufficient.

### 4.2. Incidence of AKI

In our single-center study, almost two-thirds of patients developed AKI stage 2/3 or required renal replacement therapy (RRT). The rate of AKI is comparable to other studies, whereas the proportion of patients with RRT is slightly higher. A recently published meta-analysis including 41 cohort studies, with a total of 10,282 adult patients receiving ECMO, showed an incidence of AKI and severe AKI requiring RRT of 62.8% and 44.9% [4]. The indication for dialysis was not uniformly defined in most of the studies, nor in our study, and was frequently dependent on the discretion of the treating nephrologists or intensivists. The early initiation of RRT might prevent complications from acidemia, uremia, fluid overload, and systemic inflammation, and may potentially translate into improved survival and an earlier recovery of kidney function. In addition, very restrictive fluid management with negative fluid balance—if possible—is an important therapeutic concept in the treatment of ARDS patients. According to a recent survey, providers reported that the primary indication for RRT during ECMO therapy is for either fluid overload prevention or active volume management, in nearly 60% of cases [11]. Therefore, we initiate RRT rather liberally in ARDS patients on ECMO, to prevent or treat fluid overload. This might explain our high incidence of RRT.

### 4.3. Impact of AKI

The relationship between AKI, RRT, and survival in critically ill adult patients receiving ECMO is not well defined. Thongprayoon reports that patients who develop AKI, requiring RRT while on ECMO, bear a 3.7-fold higher hospital mortality [4]. However, in other studies, AKI or the use of RRT was not associated with increased mortality, even after an adjustment for confounders. Our study confirms the results of most published studies, showing a clear association with AKI and mortality, as well as with some complications, e.g., bleedings and infections [7,8,9,10]. Due to the retrospective nature of our study, we cannot draw any causal relationship between kidney injury and other outcomes.

### 4.4. Risk Factors for AKI

The identification of risk factors for AKI and the development of a risk score in ECMO patients seems to be significant for two reasons: first, if AKI is recognized early, or the risk for AKI is even predicted, nephroprotective measures could be considered to reduce exposure to renal insults and potentially avoid the development of higher stages of AKI and reduce the associated mortality. Although the discussion about preventive or therapeutic interventions in critically ill patients with AKI is controversial, there are some strategies that may be beneficial in the ICU setting, e.g., the optimization of fluid balance and hemodynamics, as well as a medication review with the avoidance of nephrotoxic drugs, can reduce the incidence and severity of AKI and improve long-term outcomes [12]. As mentioned above, the early recognition of AKI might offer the opportunity to initiate RRT as a preventive rather than therapeutic approach.

In addition, the pathophysiology of AKI in ECMO patients is complex and multifactorial, time-dependent, and likely to be synergistic but not fully understood [13]. It appears that pre-ECMO factors, as well as ECMO-related factors, play a significant role. Pre-ECMO factors may include prolonged pre-hypoxemia and hypercapnia, hypoperfusion, ischemia–reperfusion injury, pre-existing conditions including underlying CKD, diabetes, hypertension, renovascular disease, or ongoing fluid overload with congestion. In addition, as in other extracorporeal treatments, the blood shear stress, the exposure to non-self surfaces, and the air/blood interface in ECMO patients may cause a hypercoagulable state, as well as systemic inflammation. Describing pre- as well as peri-ECMO risk factors for AKI might help to understand the pathophysiological link between ARDS, ECMO, and the kidneys.

### 4.5. Serum Creatinine

The majority of published studies in critically ill patients showed that basal renal dysfunction is a major risk factor for AKI [14]. Our study confirmed this observation. Patients with baseline renal dysfunction may have poor renal reserve function, as well as low compensatory ability.

### 4.6. Thrombocytopenia

Thrombocytopenia is a common finding among critically ill patients and is associated with an increased risk of adverse outcomes [15]. ECMO therapy has been shown to negatively affect platelet numbers and function over time [16]. Therefore, most of the available literature focuses on the platelet count under ECMO therapy and its impact on the outcome. However, in our study, a low pre-ECMO platelet count was associated with AKI. The rationale behind the relationship between initial thrombocytopenia and subsequent AKI is complex and is not fully understood [15]. Platelet count has been purported to be a marker for disease severity during acute illnesses. As inflammation/infection worsens, the consumption of platelets and thrombocytopenia may increase; it may also be compounded by local sequestration, disseminated intravascular coagulation, and endothelial activation, leading to a further decline in platelet numbers [15]. In addition, thrombocytopenia also confers a state of bleeding diathesis and increases the risk of major hemorrhage. As hemorrhagic complications are common during ECMO treatment, low pre-ECMO platelets may be associated with a higher risk of bleeding during ECMO and, thus, may lead to bleeding-associated complications, including hemodynamic instability and subsequent AKI.

### 4.7. Initial Pump Speed

In accordance with the study from Lee et al., we observed an association of initial ECMO pump speed but no correlation of the blood flow with AKI [10]. Whereas the blood flow might be a surrogate marker of severity of ARDS, as a high blow flow is indicated in cases of severe hypoxemia, the role of the pump speed in the development of AKI is not clear. An excessive ECMO pump speed may induce hemolysis and complement activation and was associated with a number of other adverse clinical outcomes in recently published studies [17]. As hemolysis has been reported to be associated with AKI in pediatric ECMO patients, this might at least partly explain our observations [18]. Therefore, clinicians must be careful not to increase the ECMO pump speed if an adequate blood flow has been accomplished.

### 4.8. Hematocrit

In our study, a low pre-ECMO hematocrit was associated with an increased risk of AKI. Studies that were mainly performed in patients undergoing cardiac surgery clearly identified the hematocrit level as a risk factor for AKI and operative mortality [19]. Accordingly, the hematocrit is one of nine parameters in the recently published SEA-MAKE, predicting major adverse kidney events after AKI in ICU patients [20]. One explanation of this association is the decreased oxygen delivery with anemia. Previous works suggested a critical threshold for oxygen delivery that may be associated with an increase in the risk of AKI. In our analyses, we did not include oxygen delivery or surrogate markers of this issue, e.g., central venous saturation.

### 4.9. Lactate

Lactate is a parameter of global tissue hypoperfusion due to inadequate oxygen delivery, resulting in tissue hypoxia and causing anaerobic glycolysis. The prognostic value of multiple types of lactate levels or calculations from lactate levels has been investigated in diverse groups of critically ill patients, for example, initial lactate levels at presentation, the duration of hyperlactatemia, and lactate clearance. There are a few studies showing that lactate levels can also be used for predicting mortality in ECMO patients [21]. In contrast, only a few recent studies suggested that lactate levels might be linked to the development of AKI. Based on our results, early lactate measurements before ECMO implantation might help to identify patients who will develop AKI in due course.

### 4.10. Base Excess

A low base excess might be a marker of systemic hypoperfusion and, therefore, a marker of illness severity. In addition, metabolic acidosis can be caused by an impairment of renal function. Studies in non-ECMO patients could show that an initial low BE is predictive of AKI in the following course of illness [22].

### 4.11. Comparison between AKI Survivors and Non-Survivors

The duration of mechanical ventilation prior to ECMO was longer in the non-survivor group. This is in accordance with a growing body of evidence, derived from many studies, suggesting that the greater the number of days of mechanical ventilation before ECMO initiation, the poorer is the outcome [23]. The RESP score and the PRESERVE score demonstrated that 7 days of mechanical ventilation before ECMO was a time point beyond which there is a reduction in survival [24,25].

In the last few years, mortality risk models have been validated to predict outcomes in this group of patients, in order to help clinicians to select appropriate candidates for ECMO treatment: the RESP score and PRESERVE score, taking into account lung and extra-lung failure, the Roch score, and ECMO net score, consider exclusively extrapulmonary organ function [24,25]. In accordance with these scores, the serum creatinine level as a marker of kidney function before ECMO initiation was higher in non-survivors than in survivors.

If there is any causative relationship between high paCO_2_ levels or primary hypercapnic acidosis and the outcome in AKI patients after ECMO initiation, it is only speculative. A retrospective multicenter cohort study may show that a PaCO_2_ of >60 mmHg at 24 h before ECMO initiation was independently associated with lower odds of ECMO weaning [26]. Thus, a high paCO_2_ level is a surrogate marker of the severity of pulmonary damage.

In addition, in a recently published study, a large relative decrease in paCO_2_ in the first 24 h after ECMO initiation was independently associated with a poor outcome in patients receiving ECMO for respiratory failure [27].

Together with a higher bicarbonate level, it could be hypothesized that hypercapnia did not occur as an acute onset in these patients but as an acute-to-chronic event with metabolic compensation. Thus, these patients might suffer from chronic lung disease with acute exacerbation, and demonstrate higher mortality compared to healthy individuals with acute lung injury.

At first glance, it seems to be surprising that the incidence of neurological complications was higher in patients with AKI who survived their hospital stay. However, looking at the data, most AKI non-survivors died while on ECMO support. The majority of these critically ill patients received deep sedation, making a reliable neurological assessment impossible. Therefore, this observation might be explained by the bias that the probability of diagnosing existing neurological disorders was different between groups.

### 4.12. Limitations

The current study suffers from some limitations. First, this study is a retrospective cohort study; however, the variables before ECMO insertion were retrieved thoroughly, with a missing rate of less than 10%. The limited sample size of our study did not allow for extensive adjustments in multivariate modeling by which to assess the predictors of mortality in AKI patients (AKI survivors vs. AKI non-survivors).

The indications for dialysis were not uniformly defined in our study, and it was frequently dependent on the discretion of the treating nephrologists or intensivists. Looking at the predictors of AKI identified in this study, it might be suggested that lactate, BE, low platelets, and impaired liver function might be markers or predictors of sepsis and, thus, not prognostic factors of AKI. In addition, infectious complications under ECMO therapy were higher in patients with AKI. However, although sepsis is the most common contributing factor for developing AKI, an AKI of any origin is associated with a higher risk of developing sepsis in the further clinical course. In a recently published study, 28% of patients had sepsis before AKI. Interestingly, 56% of patients who were not septic at the time of AKI diagnosis subsequently developed sepsis later in the hospital course [28]. Therefore, the higher incidence of infectious complications in the AKI group does not necessarily mean that patients first developed sepsis indicated/prognosticated by lactate, BE, thrombocytes, etc., and then septic AKI, but perhaps developed AKI first, followed by sepsis. Looking at the incidence of pneumonia, WBC, or CRP before ECMO implantation, there was no difference between patients with AKI and those without, suggesting that the difference in lactate, BE, etc., cannot be explained purely by an infectious status pre-ECMO, leading to sepsis and AKI under ECMO support.

The observations in this study were made over 9 years, with different ECMO types, which might influence our results. However, there was no statistically significant difference in the incidence of AKI between the different ECMO systems, nor between the study years, suggesting that there is no association between the ECMO system and AKI.

## 5. Conclusion

Severe AKI is frequently observed in patients with ARDS and ECMO therapy and is associated with increased mortality. A high lactate, serum creatinine, and ECMO pump speed, low platelets, a low base excess and a low hematocrit before ECMO initiation were independent predictors of AKI.

## Figures and Tables

**Figure 1 jcm-11-01079-f001:**
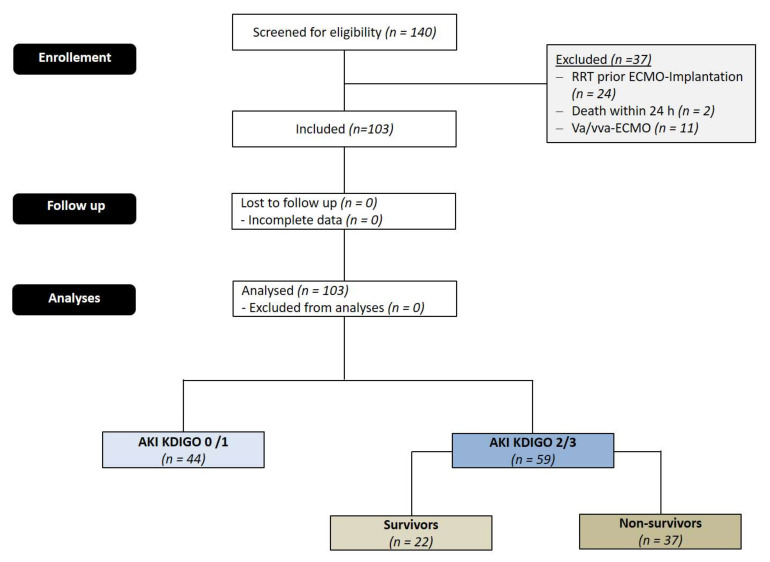
CONSORT Flow Diagram.

**Figure 2 jcm-11-01079-f002:**
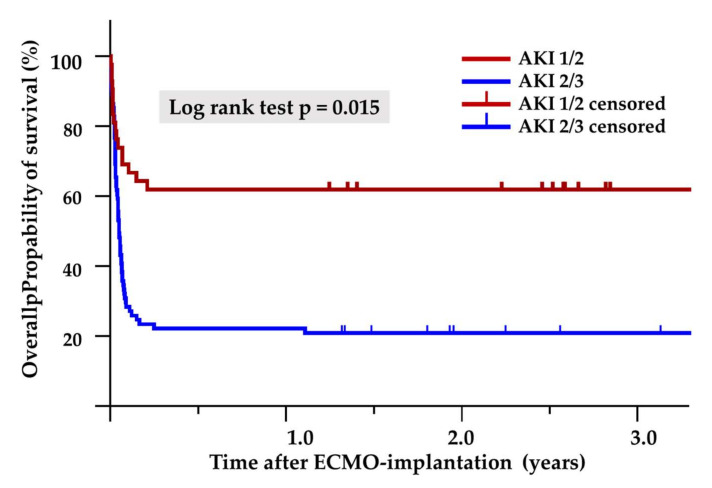
The Kaplan–Meier survival curve.

**Table 1 jcm-11-01079-t001:** Patients’ clinical and laboratory pre-ECMO characteristics, between AKI and non-AKI patients.

	AKI 0/1 (*n* = 44)	AKI 2/3 (*n* = 59)	*p*-Value
Age (years)	53.35 ± 16.56	55.19 ± 14.87	n.s.
Height (cm)	172.47 ± 7.12	171.47 ± 10.23	n.s.
Weight (kg)	83.08 ± 23.63	84.15 ± 20.36	n.s.
BMI (kg/m^2^)	24.17 ± 6.45	24.45 ± 5.29	n.s.
Female gender	16 (36.4)	22 (37.3)	n.s.
Reason for ARDS (*n*, %)			
Pneumonia	27 (61.4)	42 (71.2)	n.s.
Exacerbation of COPD	5 (11.4)	3 (5.8)	n.s.
H1N1 Pneumonia	4 (9.1)	6 (10.2)	n.s.
Other	8 (18.2)	7 (11.9)	n.s.
Primary oxygenation failure (*n*, %)	33 (75.0)	47 (79.7)	n.s.
Primary hypercapnic acidosis (*n*, %)	11 (25.0)	12 (20.3)	n.s.
aHTN (*n*, %)	24 (54.5)	32 (54.2)	n.s.
AFib (*n*, %)	5 (11.4)	8 (13.6)	n.s.
PHT (*n*, %)	4 (9.1)	8 (13.6)	n.s.
Diabetes (*n*, %)	4 (9.1)	15 (25.9)	0.043
CAD (*n*, %)	12 (27.3)	14 (23.7)	n.s.
s/p CPR (*n*, %)	6 (13.6)	6 (10.2)	n.s.
HIT (*n*, %)	3 (6.8)	0 (0)	n.s.
Liver failure (*n*, %)	1 (2.3)	5 (8.5)	n.s.
CKD (*n*, %)	1 (2.3)	15 (25.4)	<0.001
Duration of mechanical ventilation (h)	85.12 ±134.70	87.54 ± 100.55	n.s.
FiO_2_	91.00 ± 17.87	93.53 ± 13.97	n.s.
P_insp_ (mbar)	33.77 ± 5.72	33.16 ± 5.43	n.s.
PEEP (mbar)	14.56 ± 4.40	14.33 ± 4.24	n.s.
PaO_2_/FiO_2_ (mmHg)	10.55 ± 72.82	92.01 ± 58.92	n.s.
Tidal volume (mL)	414.07 ± 131.92	438.88 ± 136.57	n.s.
Compliance (mL/mbar)	23.45 ± 10.02	24.93 ± 11.29	n.s.
LISS	3.52 ± 0.55	3.51 ± 0.50	n.s.
paO_2_ (mmHg)	91.31 ± 53.27	78.63 ± 29.33	n.s.
pCO_2_ (mmHg)	79.50 ± 31.33	72.01 ± 24.92	n.s.
pH	7.23 ± 0.13	7.20 ± 0.13	n.s.
Base excess (mmol/L)	2.36 ± 5.43	−1.70 ± 7.56	0.007
HCO_3_ (mmol/L)	31.37 ± 7.22	26.74 ± 7.70	0.006
SaO_2_ (%)	92.49 ± 5.77	90.64 ± 7.14	n.s.
Lactate (mmol/L)	1.58 ± 1.29	2.73 ± 2.40	0.008
Hemoglobin (g/dL)	11.49 ± 2.61	10.69 ± 1.79	n.s.
Hematocrit (%)	35.39 ± 7.06	32.35 ± 5.10	0.017
Urea (mmol/L)	7.85 ± 5.20	13.08 ± 8.43	0.001
Serum creatinine (mmol/L)	83.19 ± 30.11	144.94 ± 95.95	0.001
WBC (×10^9^/L)	15.33 ± 7.94	14.76 ± 9.03	n.s.
Procalcitonin (µg/L)	4.31 ± 7.71	13.69 ± 28.51	n.s.
CRP (mg/L)	202.37 ± 136.33	199.89 ± 127.97	n.s.
INR	1.17 ± 0.23	2.01 ± 0.19	n.s.
aPTT (s)	41.21 ± 21.18	48.12 ± 20.99	n.s.
D-dimers (mg/L)	8.92 ± 12.06	7.54 ±10.49	n.s.
Initial blood flow ECMO (L/min)	3.38 ± 0.89	3.69 ± 0.822	n.s.
Initial revolution (min^−1^)	2970.55 ± 348.33	3370.53 ± 472.59	0.035
Initial airflow (L/min)	3.50 ± 1.56	3.64 ± 1.55	n.s.
Platelets (×10^9^/L)	288.57 ± 186.39	185.21 ± 137.92	0.002
CK (U/L)	423.47 ± 938.25	560.46 ± 1008.46	n.s.
Total bilirubin (µmmol/L)	16.89 ± 22.71	28.14 ± 48.86	n.s.
AST (U/L)	83.53 ± 126.49	86.07 ± 88.13	n.s.
ALT/(U/L)	53.74 ± 65.53	51.72 ± 72.95	n.s.
LDH (U/min)	393.12 ±197.70	438.11 ± 254.44	n.s.
Albumin (g/dL)	24.45 ± 7.97	23.01 ± 7.31	n.s.

AKI = acute kidney injury, AKI 0/1 = no acute kidney injury or acute injury stage 1 according to the Kidney Disease: Improving Global Outcomes classification, AKI 2/3 = acute injury stage 2 or 3 according to the Kidney Disease: Improving Global Outcomes classification, Afib = Atrial fibrillation, aHTN = arterial hypertension, ALT = aspartate aminotransferase, aPTT = activated partial thromboplastin time, ARDS = acute respiratory distress syndrome, AST = alanine transaminase, BMI = body mass index, CAD = coronary artery disease, CK= creatinine kinase, CKD = chronic kidney disease, COPD = chronic obstructive pulmonary disease, CPR = cardiopulmonary resuscitation, CRP = C-reactive protein, HIT = heparin-induced thrombocytopenia, INR = international normalized ratio, LDH = lactate dehydrogenase, LISS = lung injury severity score, n.s. = not sifgificant, WBC = white blood cell count.

**Table 2 jcm-11-01079-t002:** The outcomes of patients, comparing AKI 2/3 and AKI 0/1 groups.

	AKI 0/1 (*n* = 44)	AKI 2/3 (*n* = 59)	*p*-Value
GI bleeding (*n*, %)	5 (11.4)	12 (20.3)	n.s.
ENT bleeding (*n*, %)	22 (50.0)	47 (79.7)	0.008
ECMO bleeding (*n*, %)	16 (36.4)	31 (52.5)	n.s.
Pulmonary bleeding (*n*, %)	5 (11.4)	17 (28.8)	0.033
Hemolysis (*n*, %)	1 (2.3)	5 (8.5)	n.s.
Pneumothorax (*n*, %)	7 (15.9)	6 (10.2)	n.s.
Cardiac complications (*n*, %)	17 (38.6)	19 (32.2)	n.s.
Neurological complications (*n*, %)	11 (25)	19 (32.2)	n.s.
Infectious complications (*n*, %)	35 (79.5)	56 (94.9)	<0.001
Blood stream infection (*n*, %)	12 (27.2)	33 (55.9)	0.005
CPR (*n*, %)	4 (9.1)	3 (5.1)	n.s.
Weaning failure from ECMO (*n*, %)	15 (34.1)	30 (50.9)	0.08
30-day mortality (*n*, %)	17 (38.6)	35 (59.3)	n.s.
Hospital mortality (*n*, %)	17 (38.6)	37 (62.7)	0.021

ENT = ear, nose, and throat, ECMO = extracorporeal membrane oxygenation, CPR = cardiopulmonary resuscitation, GI = gastrointestinal.

**Table 3 jcm-11-01079-t003:** Predictors of AKI 2/3 in patients undergoing ECMO implantation.

	OR	95% CI	*p*-Value
Lactate pre-implant	1.42	1.033–1.955	0.002
Serum creatinine pre-implant	1.018	1.007–1.003	0.001
Thrombocytes pre-implant	0.995	0.992–0.999	0.006
Hematocrit pre-implant	0.917	0.852–0.988	0.022
Initial Pump speed	1.009	1.001–1.005	0.046
Base excess pre-implant	0.913	0.837–0.967	0.001

**Table 4 jcm-11-01079-t004:** Patients’ clinical and laboratory pre-ECMO characteristics, between AKI survivors and AKI non-survivors.

	AKI Survivors (*n* = 22)	AKI Non-Survivors (*n* = 37)	*p*-Value
Age (years)	52.38 ± 15.80	56.79 ± 14.29	n.s.
Height (cm)	173.47 ± 8.20	170.20 ± 11.28	n.s.
Weight (kg)	81.18 ± 16.89	86.03 ± 22.36	n.s.
BMI (kg/m^2^)	23.44 ± 16.89	86.03 ± 22.36	n.s.
Female gender	5 (22.7)	17 (45.9)	n.s.
Reason for ARDS (*n*, %)			
Pneumonia	14 (63.6)	28 (75.7)	n.s.
Exacerbation of COPD	2 (9.1)	1 (2.7)	n.s.
H1N1 Pneumonia	4 (18.2)	2 (5.4)	n.s.
Others	2 (9.1)	5 (13.5)	n.s.
Primary hypercapnic acidosis (*n*, %)	0 (0.0)	12 (32.4)	0.003
Primary oxygenation failure (*n*, %)	22 (0.0)	25 (67.6)	0.003
aHTN (*n*, %)	10 (45.5)	22 (59.5)	n.s.
AFib (*n*, %)	3 (13.6)	5 (13.5)	n.s.
PHT (*n*, %)	1 (4.5)	7 (18.9)	n.s.
Diabetes (*n*, %)	8 (36.4)	7 (18.9)	n.s.
CAD (*n*, %)	5 (22.7)	9 (24.3)	n.s.
s/p CPR (*n*, %)	3 (13.6)	3 (8.1)	n.s.
HIT (*n*,%)	0 (0)	0 (0)	n.s.
Liver failure (*n*, %)	1 (4.5)	4 (10.8)	n.s.
CKD (*n*, %)	4 (18.2)	11 (50.0)	n.s.
Duration of mechanical ventilation (h)	75.90 ± 86.81	91.14 ± 108.16	0.078
FiO_2_	98.05 ± 4.58	91.06 ± 16.6	n.s.
P_insp_ (mbar)	33.61 ± 4.99	32.92 ± 5.70	n.s.
PEEP (mbar)	14.77 ± 3.78	14.11 ± 4.50	n.s.
PaO_2_/FiO_2_ (mmHg)	77.16 ± 29.24	99.42 ± 68.43	n.s.
Tidal volume (mL)	444.92 ± 95.33	434.95 ± 160.10	n.s.
Compliance (mL/mbar)	23.69± 7.27	21.73 ± 7.37	n.s.
LISS	3,61 0.39	3.44 ± 0.55	n.s.
paO_2_ (mmHg)	75.77 ± 28.70	80.13 ± 27.9	n.s.
pCO_2_ (mmHg)	64.44 ± 44.31	76.12 ± 27.90	0.08
pH	7.20 0.12	7.19 ± 0.15	n.s
Base excess (mmol/L)	−4.23 ± 9.8	0.52 ±8.3	0.1
HCO_3_ (mmol/L)	23.36 ± 4.19	28.22 ± 8.44	0.04
SaO_2_ (%)	89.38 ± 0.37	91.28 ± 6.61	n.s.
Lactate (mmol/L)	3.48 ± 2.35	2.31 ± 2.36	n.s.
Hemoglobin (g/dL)	10.92 ± 2.15	10.57 ± 1.57	n.s.
Hematocrit (%)	32.63 ± 6.54	32.19 ± 4.19	n.s.
Urea (mmol/L)	13.63 ± 6.54	13.08 ± 9.19	n.s.
Serum creatinine (mmol/L)	124.95 ± 77.77	180.92 ± 115.72	0.03
WBC (×10^9^/L)	15.33 ± 9.10	14.44 ± 9.11	n.s.
Procalcitonin (µg/L)	22.95 ± 36.13	8.32 ± 22.3	n.s.
CRP (mg/L)	212.98 ± 130.90	192.41 ± 127.57	n.s.
INR	1.25 ± 0.19	1.23 ±0.30	n.s.
aPTT (s)	47.55 ± 18.49	48.44 ± 22.51	n.s.
D-dimers (mg/L)	4.39 ± 5.56	9.48 ± 12.4	n.s.
Initial blood flow ECMO (L/min)	3.43 ± 0.72	3.83 ± 0.54	n.s.
Initial revolution (min^−1^)	3193.75 ± 379.89	3424.92 ± 498.08	n.s.
Initial airflow (L/min)	3.58 ± 1.23	3.66 ± 1.73	n.s.
Platelets (×10^9^/L)	168.75 ±170.84	194.36 ± 117.51	n.s.
CK (U/L)	380.21 ± 586.14	678.55 ± 1204.05	n.s.
Total bilirubin (µmmol/L)	23.97 ± 24.7	30.48 ± 58.30	n.s.
AST (U/L)	67.59 ± 55.57	97.99 ± 103.3	n.s.
ALT/(U/L)	68.45 ± 209.67	43.09 ± 43.67	n.s.
LDH (U/min)	501.88 ± 361.69	402.68 ± 167.33	n.s.
Albumin (g/dL)	21.65 ± 7.67	23.77 ± 7.16	n.s.

AKI = acute injury stage 2 or 3 according to the Kidney Disease: Improving Global Outcomes classification, Afib = Atrial fibrillation, aHTN = arterial hypertension, ALT = aspartate aminotransferase, aPTT = activated partial thromboplastin time, ARDS = acute respiratory distress syndrome, AST = alanine transaminase, BMI = body mass index, CAD = coronary artery disease, CK= creatinine kinase, CKD = chronic kidney disease, COPD = chronic obstructive pulmonary disease, CPR = cardiopulmonary resuscitation, CRP = C-reactive protein, HIT = heparin-induced thrombocytopenia, INR = international normalized ratio, LDH = lactate dehydrogenase, LISS = lung injury severity score, n.s. = not sifgificant, WBC = white blood cell count.

**Table 5 jcm-11-01079-t005:** The outcomes of patients, comparing AKI-survivors and AKI-non-survivors.

	AKI-Survivors (*n* = 22)	AKI-Non-Survivors (*n* = 37)	*p*-Value
GI bleeding (*n*, %)	5 (22.7)	7 (18.9)	n.s.
ENT bleeding (*n*, %)	16 (72.7)	31 (83.8)	n.s.
ECMO bleeding (*n*, %)	10 (45.5)	22 (59.5)	n.s.
Pulmonary bleeding (*n*, %)	4 (18.2)	13 (35.1)	n.s.
Hemolysis (*n*, %)	2 (9.1)	4 (10.8)	n.s.
Pneumothorax (*n*, %)	3 (18.2)	3 (8.2)	n.s.
Cardiac complications (*n*, %)	3 (13.6)	16 (43.2)	0.0186
Neurological complications (*n*, %)	11 (50.0)	8 (21.6)	0.024
Infectious complications (*n*, %)	19 (86.4)	37 (100.0)	0.047
Blood stream infection (*n*, %)	12 (54.5)	21 (56.8)	n.s.
CPR (*n*, %)	0 (0)	3 (8.0)	n.s.
Weaning failure from ECMO (*n*, %)	0 (0)	30 (81.1)	0.0001

ENT = ear, nose, and throat, ECMO = extracorporeal membrane oxygenation, CPR = cardiopulmonary resuscitation, GI = gastrointestinal.

## Data Availability

The data presented in this study are available on request from the corresponding author. The data are not publicly available due to legal or privacy issues.

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
