# Peer review of "Acute Kidney Injury in Patients with Severe ARDS Requiring Extracorporeal Membrane Oxygenation: Incidence, Prognostic Impact and Risk Factors"

_jcm, 2022, doi:10.3390/jcm11041079_

Round 1
Reviewer 1 Report
Comments to the Authors,
 The present study was designed to define the incidence, clinical course as well as predictors of AKI in adults receiving ECMO support. Although this is a monocentric analysis, sample size was not so small and obtained results were clearly presented. In addition, authors describe manuscript clearly, and discussion based on the obtained results and previous reports seemed not be contradiction. However, novelty of the present study appeared to be low, described data were not impressive and within the presumption for the physicians, Thus, could you please show additional data from another angle that increase the novelty and fascination of the present manuscript. Concretely, please show the data and comment regarding the crucial difference between the patients with severe AKI who survived and those who died under the support with ECMO.
 Collectively, the present study will not reach the level for acceptance because of the lack of novelty and attractiveness although this manuscript did not contain contradiction and inappropriate data.
Author Response
Dear Reviewer,
Thank you very much for your extremely fast review and your helpful comments to increase the quality of our manuscript. Below, please find our reply:
We are happy to include the data about AKI-survivors compared to AKI-non-survivors in our revised manuscript.
Results: A primary hypercapnic acidosis was more common in AKI-non-survivors (12 (32.4%) vs. 0 (0.0%), p < 0.01). Accordingly, pCO2-levels prior to ECMO-implantation tended to be higher in AKI-non-survivors (76.12 ± 27.90 mmHg vs. 64.44 ± 44.31 mmHg, p=0.08) In addition, duration of mechanical ventilation prior to ECMO-implantation tended to be longer (91.14 ± 108.16 h vs. 75.90 ± 86.81 h, p = 0.078), serum-creatinine (180.92 ± 115.72 mmol/l vs. 124.95 ± 77.77 mmol/l, p =0.03) and bicarbonate levels were significantly higher in non-survivors (28.22 ± 8.44 mmol/l vs. 23.36 ± 4.19 mmol/l, p =0.04). Whereas cardiac complications (16 (43.2%) vs. 3 (13.6%), p = 0.0186) as well as infectious complications (37 (100.0%) vs. 19 (86.4%), p =0.047) could be observed more frequently in AKI-non-survivors, incidence of neurological complications was significantly higher in AKI-survivors (11 (50.0%) vs. 8 (21.6%) p = 0.024). Out of 37 AKI-non-survivors, seven patients could be weaned successfully from ECMO, whereas 30 patients died during extra-corporeal support.
Comment: Duration of mechanical ventilation prior to ECMO was longer in non-survivor-group. This is in accordance with a growing body of evidence derived from many studies that suggest that the greater number of days of mechanical ventilation before ECMO initiation, the poorer is the outcome [25]. The RESP score and the PRESERVE score demonstrated 7 days of mechanical ventilation before ECMO as a time point beyond which there is a reduction in survival [1,2].
In the last years, mortality risk models have been validated to predict outcome in this group of patients in order to help clinicians to select appropriate candidates for ECMO treatment: RESP-score and PRESERVE-score, take into account lung and extra lung failure, Roch-score and ECMO-net score consider exclusively extrapulmonary organ function [2,3]. In accordance with these scores, serum creatinine as a marker of kidney function before ECMO-initiation was higher in-non-survivors than in survivors.
If there is any causative relationship between high paCO2-levels or rimary hypercapnic acidosis and outcome in AKI-patients after ECMO-intiation is speculative. A retrospective multicenter cohort study could show that a PaCO2 >60 mmHg at 24 h before ECMO initia-tion was independently associated with lower odds of ECMO weaning [4]. Thus, a high paCO2 is a surrogate marker of severity of pulmonary damage.
In addition, a large relative decrease in PaCO2 in the first 24 hours after ECMO initiation was independently associated with poor outcome in patients receiving ECMO for respira-tory failure in a recently published study [5].
Together with a higher bicarbonate level it could be hypothesized that hypercapnia did not occur as acute-onset in these patients but as acute-to-chronic-event with metabolic compensation. Thus, these patients might suffer from chronic lung disease with acute ex-acerbation with a higher mortality compared to healthy individuals with acute lung injury.
At first glance, it seems to be surprising that the incidence of neurological complications was higher in patients with AKI that survived their hospital stay. However, looking at the data most AKI-non-survivors died on ECMO-support. The majority of these critically ill patients received deep sedation making a reliable neurological assessment impossible. Therefore, this observation might be explained by the bias that the probability of diagnosing existing neurological disorders was different between groups. Please find atteched the revised manuscript.

Reviewer 2 Report
This paper is a nice presentation of the clinical problems in patients undergoing ECMO. Renal replacement therapy ist now one of the essential tools in the ICU, and in the therapeutic setting its use enables and expands clinical options and is not a sign of poor patient prognosis.
Author Response
Dear Reviewer,
thank you very much for your fast and positive comments.
Round 2
Reviewer 1 Report
Comments to the Authors and Editor, In the revised manuscript, it was shown that authors made an effort to improve the novelty and attractiveness. I am so satisfied with new results regarding the difference between the patients with severe AKI who survived and those who died under the support with ECMO. Additional new results dramatically improved and increased the quality of manuscript, so the revised manuscript could reach the level for acceptance. Sincerely yours,